# The changing meaning of 'home' in the work of South African women academics during the pandemic-enforced lockdown

Cyrill Walters[1]*, Linda Ronnie[2], Jonathan Jansen[3], Samantha Kriger[4]

1 Stellenbosch Business School, Stellenbosch University, Stellenbosch, South Africa, 2 School of Management Studies, University of Cape Town, Cape Town, South Africa, 3 Department of Education Policy Studies, Stellenbosch University, Stellenbosch, South Africa, 4 Faculty of Education, Cape Peninsula University of Technology, Cape Town, South Africa

* cyrillwalters@sun.ac.za

**Data Availability Statement:** The research team will share data upon request due to ethical restrictions on sharing the data publicly. The dataset contains potentially identifying information i.e., institution, career stage, number of children as

## Abstract

This article shows how the meaning of home and 'working from home' were fundamentally transformed by the pandemic-enforced lockdown for women academics. Drawing on the experiences of more than 2,000 women academics, we show how the enduring concept of *home* as a place of refuge from the outside world was replaced with a new and still unsettled notion of home as a gendered space that is a congested, competitive, and constrained setting for women's academic work. In this emerging new place for living and working, home becomes a space that is claimed, conceded, and constantly negotiated between women academics and their partners as well as the children and other occupants under the same roof. Now, as before, home remains a deeply unequal place for women's work, with dire consequences for academic careers. It is therefore incumbent upon women academics and higher education institutions to develop a deep understanding of the social meanings of home for academics, and the implications for the 'new normal' of working from home.

## Introduction

Over the years, poets, architects, designers, and scholars have all weighed in on the many meanings of home [1–4]. What becomes clear from both academic and professional writing on the subject is that home can be conceived of as both a space and a place [5]. With its fixed size, layout, and features, the home is the physical *space* that hosts our everyday activities. Yet, it is much more than a physical structure occupied by individuals or families. As the environment where we think, feel, and experience in the abstract, the home is the *place* that gives us comfort and peace. In much of the literature, a persistent theme is that of home as a place of refuge from the outside world. It is where you can 'put your feet up'. Home, in this sense, is a private space and a private place, where the dweller can regulate social interaction with the outside world. "The psychological need for privacy," argues Despres [6] in her well-cited review of the literature on the meaning of home, "is among the most powerful theoretical concepts that have been used to explain the meaning of the home as a refuge" (p. 102). The house therefore acts as a fortress or enclosure which sets a boundary between the inside and outside worlds, so that "when the boundary is transgressed, people feel invaded in their privacy" (ibid).

well as several demographic data participants have shared in the open-ended section of the survey. These restrictions have been imposed by the university's Research Ethics Committee (REC): Social, Behavioral and Education Research (SBE). Given its sensitivity, the data has been anonymized, and a nonidentifying unique ID given to each female academic. Thus, the participants' privacy is protected, and the data's utility for reanalysis or use also preserved. For these reasons, the data will only be accessed by the research team, and if required, be made available to others under an appropriate and approved data sharing agreement. Ethical clearance for this study was granted by: REC: Social, Behavioral and Education Research (SBER) Stellenbosch University Project number: REC-2020-15216 The REC conditions clearly state that only the research team will have sole access to the data and that the data cannot be publicly shared. However, upon request to the Principal Investigator of this study Professor Jonathan Jasen (Jonathan.jansen@sun.ac.za) data sets will be made available fully de-identified or anonymized from all demographic indicators. If you have any questions regarding the ethics application or the conditions set, please contact: Clarissa Graham at cgraham@sun.ac.za who is the REC Secretariat: Research Ethics Committee: Social, Behavioral and Education Research (REC: SBE).

**Funding:** The authors received no specific funding for this work.

**Competing interests:** The authors have declared that no competing interests exist.

The advent of telecommuting—when employees started to work away from the traditional office by means of communications devices connecting them to their job—naturally changed this notion of home as a refuge from the external world. For example, a 2014 study of academics at a distance-education university in South Africa described the disruptive effects of remote work on home life, including family-work role conflicts and overwork [7, 8]. However, the COVID-19 pandemic and the subsequent lockdown mandates cast the opportunities and challenges of 'work from home' in an entirely different light. In South Africa, as in many parts of the world, the state response to COVID-19 included a country-wide lockdown that lasted in various stages of severity from 27 March 2020 to 21 September 2020. At the onset, at the strictest lockdown stage ("level 5"), all schools and non-essential businesses were closed, employees were required to work from home where possible, and mobility outside the home was significantly restricted. Almost overnight, the meaning of the home "lost its meaning as a start and end point of the daily cycle" and "gained a new meaning as the only place in our everyday lives during self-quarantine" [9].

Already, research is emerging to show how variables such as gender, parental status, and profession impact how the home is experienced by people during lockdown. Female academics present an interesting subset given the availability of institutionalised measures of professional outputs in the form of research publications and the orientation of their field toward knowledge gathering and dissemination. In 2020, early statistical accounts emerged exploring women's scientific productivity during the pandemic [10, 11], and how their publishing rates appear to be falling relative to their male peers [12–14], particularly for women with children [15]. Other studies showing the adverse impacts of the pandemic on women and mothers more generally provide greater context for these disparities: increase in household and childcare workload [16]; greater work disruption [17] and reduction to work hours [18] due to increased childcare responsibilities; higher levels of stress [19, 20]; worse sleep quality [21]; and more symptoms of mental health disorders including depression, anxiety, and post-traumatic stress disorder [22]. Indeed, accounts such as that by Guy and Arthur [23] in the U.S. show in great depth the guilt, anxiety, and frustration of women who are struggling to balance their dual roles as academics and mothers while confined to the home. Meanwhile, other accounts of single academics with no children, such as Carreri and Dordani in Italy [24], suggest that working from home during the lockdown has been a period of "serenity and introspection".

At least one study moved beyond accounts of women's academic productivity to examine how universities in Australia support academic women working from home [25]. Such positive developments are predicated on an understanding of the different ways the home is changing for women academics as a space to perform their necessary activities and a place of privacy, comfort, and refuge. As yet, however, there has been no large-scale, qualitative, and systematic research on the impact of a pandemic-enforced lockdown on academics working from home, and what this means for the changing concept of home among women scholars and scientists in an emerging literature that is so emphatically centred on the experiences of people in developed countries in the Global North [26]. This is what our study offers, from the perspective of academic women in a developing African country where economic dynamics and the 2020 pandemic response present a case that is at once global and 'extreme'.

## South Africa: A confined space

On 27 March 2020, South Africa implemented one of the most stringent lockdown responses in the world. This initial and most severe phase of the lockdown, referred to as "stage 5", included curfews, school closures, non-essential business closures, banned sales on alcohol

and tobacco, work from home mandates, and travel bans. Along with Uganda, South Africa had some of the strictest home confinement measures in sub-Saharan Africa [27]. South Africans were not allowed to leave their homes for any reason other than to purchase or produce essential goods, making it an "extreme country case" [28]. Although some of these restrictions were marginally lifted in May 2020 –with the permission of limited outdoor exercise during certain hours and sale of some non-essential goods under "stage 4" lockdown conditions–the severity of home confinement in the initial six months of the lockdown had particularly important ramifications for South Africans who were able to work from home, including academics. Although this severity would appear to make South Africa an outlier, the economic structure of the country makes the study of women academics an important data point in the study of professional working women and their relationship to home.

South Africa is regarded as a dual economy, consisting of a large, low-skilled informal economy characteristic of emerging countries, and a small, high-skilled formal economy on par with developed countries. South Africa's economic duality manifests in high rates of unemployment (29% in 2019) [29], extreme wealth inequality (a Gini coefficient of 0.63) [30], and a polarised labour market. Academics constitute a rarefied subset of the formal labour market. In 2017, when the total labour force was an estimated 22 million [31], 53,000 were employed as temporary or permanent academic staff across South Africa's 26 universities, 50% of whom were female [32]. A growing gap in rising student enrolment and stagnating academic staffing has placed increased pressure on maintaining and improving the academic pipeline. At the same time, the threat of increased turnover from academic staff who find they are insufficiently remunerated relative to their workload and the alternative opportunities available in international universities or the public or private sector, has intensified the need to retain academic talent in the country [33]. Retention of women academics is of particular import as they are underrepresented in leadership and higher-level academic positions (despite constituting roughly half of the total academic work force.

And yet, national attitudes overall are progressive about women's role in the workplace, and in the family. In a 2012 survey of social attitudes, 88% of South Africans felt that both men and women should contribute to household income and only 46% believed in traditional gender roles: men as primary money-earners and women as primary household-minders [34]. Still, the majority of employed South African women reported having the sole or usual responsibility of laundry and household cleaning (65%) and caring for sick family members (55%). And when it came to having young children under the age of five, a majority of men (51%) and women (52%) felt that the best way to organise the family was for the mother to stay home while the father worked full time. This survey presumes a traditional heteronormative family structure and does not address alternative approaches, namely same-sex partnerships. It does, however, affirm the gendered nature of domestic work in South Africa, the "double burden" on women who are employed [35].

## Home as a gendered space

As mobility became limited and spaces confined during the pandemic lockdown, and daily routines and trajectories were abruptly discontinued, questions about where people could go, which spaces were safe, and what they would do where suddenly became urgent matters of survival overnight. For so many professionals facing mandates to 'work from home', there was no more important space to consider and renegotiate than the home, now the site of their new office and work activities. That the home is conceived as a gendered space due to the traditional association of women–any woman–with domestic work is not new. Nor is teleworking, and the notion of home as a dual space for work and non-work activities an advent of the

Covid-19 era. In fact, for the past two decades there has been optimism that options to work from home would liberate workers by increasing flexibility, improving productivity, and improving work-life balance [36].

But as Mallett et al. explain, "the ways in which spaces are 'lived in', involve not only practical activities but also the user's symbolic understanding and imagination" [36]. How does the symbolic understanding of the home space change when women in a demanding professional field are confined to their homes and expected to maintain their work performance (teaching, research, and administration), at a time when their partners face similar professional demands; children, elderly, and ill family members have lost their institutional support systems; and no one else may enter the home space to assist in any way?

Following Lefebvre's [37] conception of the home as a social space, Mallett et al. [36] theorise that the gendered nature of the "domestic sphere" has profound implications for women who work within their homes. In order to construct working policies that are actually liberating for both women and men, they argue that a deep understanding of the home as a social space and its complexities must be taken into account. These complexities include the pressure to attend to childcare or housework in the home space; actual barriers in the home between a workspace and living space; positive associations with the workplace; the impacts on well-being of social isolation and spending so much time in the same space; having to share the at-home workspace with others; negative impacts on relationships with partners and children; domestic violence in the home; and socio-economic differences among workers.

Studies emerged early on in the pandemic indicating that the result, globally, was women taking up the burden of caring responsibilities, to the detriment of their work productivity and future employment prospects [38–40], and that South Africa was no exception [35]. Although some studies have shown that the sudden increased burden in domestic work led more men to take up domestic duties, there was a limit to the parity achieved [41]. In order to better understand the gendered nature of working from home as proposed by Mallett et al. [36], this study aims to look beyond metrics of productivity and workload to examine the experiential change that comes with spatial change: *How does the meaning of home change for women academics when a global crisis confines them and their work to the home*? The paper proceeds with a discussion of the methodology and data analysis, followed by a presentation of the findings and a discussion of the key insights, and concluding with a summary of the implications for future research on this topic.

## Method

This is a segmental report on a larger study of 2,029 academic women in South Africa's 26 public universities, who completed a survey questionnaire with 13 Likert-scale questions to gauge the impacts of the lockdown on the work and home lives of academic women, followed by an open-ended section with unlimited space for respondents to comment on any aspect of the impact of the lockdown on their academic work. A pilot study was conducted using a draft survey instrument, which was subsequently improved and finalised. Ethical clearance for the study was granted by the host institution (Stellenbosch University) and ethical review was obtained from gateway clearance certificates in most of the 26 universities. The distribution of the online survey varied by institution: in some cases, the research team was permitted to contact academics directly while in most cases, the institution distributed the survey link to women academics. Participation was voluntary and participants agreed electronically via tick box to an informed consent form before completing the online survey.

The online survey was opened for submissions on 1 July 2020 and closed on 30 September 2020; thus, the survey was completed within the lockdown period, although the women's

recounted experiences include the initial, more stringent "hard" lockdown ("stage 5", from 27 March) through to the eased, lower-level lockdown ("stage 2", from 18 August). During all of these stages, universities were closed to face-to-face teaching, and only minor exceptions were made, with institutional permission, for some professional fields (e.g., medical students' practicums) and laboratory work. At this time, it was estimated that there were between 24,332 and 25,857 women academics in South Africa, resulting in an average of 8.3% of South African women academics included in this study.

This segment of this study, on the meanings of home, draws on the data from the open-ended question at the end of the survey, which was completed by 1,857 participants and comprised more than 221,000 words. An interpretative, qualitative approach was taken to analyse responses related to the meaning of home. Coleman and Unrau's [42] six-step framework was used to extract responses, code extracts of the data, and consolidate themes based on the emergent themes. The qualitative meanings of encoded words and sentences were compiled and analysed using Atlas.ti software. Although the vast majority of respondents alluded to their home or a sense of place in some regard (e.g., the process of working from home or a desire to be in the office), the responses were specifically analysed for references to the home as a "space" or "environment" and descriptions of the experiential and physical changes in one's living space during the lockdown. Excerpts that met these criteria were isolated and coded for emergent themes.

Survey responses that described the changing nature of the home during the lockdown came from women academics working at a variety of institutions, with and without children, and at different career stages:

- Established (16+ years)

- Mid-career (11–15 years)

- Experienced (6–10 years)

- Early career (0–5 years)

For those with children in the home, these children ranged from infants to young adults, and included toddlers, kindergarteners, primary-school aged children, teenagers, and university students. These two aspects–parental status and career stage–were the only demographics gathered from the sample to ensure confidentiality of both respondents and their respective universities.

## Findings

Across institutions, career stage, parental status, and child age, participants expressed common experiences that transcended these variables. Overall, these participants described negative experiences of their changing home environment–a change for the worse. There were minor exceptions of positive experiences of the changing home environment. Notably, these came from established academics. From their collective descriptions, five dominant themes emerged which can be broken into two categories: challenges of the changing home environment and the impact of these challenges. These challenges and impact are congruent with the gendered aspects of homebased working proposed by Mallett et al. [36]. Table 1 summarises the frequency of the themes and their connection to the specific space-based considerations cited by Mallet et al.

## Challenges

**Blurring spaces, blurring lines.** "The boundaries between work and home are now blurred by my presence at home." These words from an established academic with no children

**Table 1. Emergent themes and their connection to gendered aspects of homebased working from Mallett et al. [36].**

| Theme | | Frequency | Aspect from Mallett et al. (2020) |
|---|---|---|---|
| Challenges | | | |
| | *Blurring spaces, blurring lines* | 385 | Increased workload [43]; Increased domestic work [44]; Struggle to separate work and home [45] |
| | *Sharing space* | 95 | Challenges of sharing space [46–49]; Household overcrowding [50] |
| | *Clashes with partners* | 92 | Reduced satisfaction with partner [51] |
| Impacts | | | |
| | *Work suffers* | 274 | Job loss for women [52]; Employment insecurity [53] |
| | *Mental and emotional health toll* | 204 | Social isolation [54]; Adverse impacts on well-being [40]; Impacts from lack of organisational support [55] |

capture the most prominent theme expressed by the participants about their changing home space. As work activities were brought into the home, they tended to overlap with, and eventually overtake, home activities. As an early-career academic and mother of two explained, "It became more of a living-at-work situation than working-from-home situation."

This blurring of the line between work and personal activity challenged the meaning of the home as a place of refuge. "It was extremely difficult to adjust to the new working environment, as it is the same environment where I used to 'relax' after work," wrote one early-career academic with no children. It was not only work that blurred boundaries for academics but the needs of children as all schools and childcare services were closed under the lockdown mandates:

> Home schooling is difficult, especially if the living room serves as classroom and office, and the television is an additional screen. The lines are blurred between what identity do I give to space or tool at certain time (sic) of the day. (Established academic with one child in primary school and another in high school)

Spatially, "the lockdown squeezed everything under one roof", as early-career mother of a kindergartner and a primary-schooler put it. This made it difficult and confusing for academics to distinguish between the different facets of their lives. As another established academic with a high school child explained, "I feel like I cannot compartmentalise work from home anymore—they blur into each other—because of the constant guilt of whether I am doing enough on all fronts." As a result, time also became blurred, which made it difficult for academics to manage their energy. "Working time extends until the evening, because you are expected to continuing replying to emails and students. Work time and home time is blurred, and it is tiring," explained an early-career woman with no children.

Although the additional household duties were one reason female academics cited for pushing them to work outside normal business hours, the scope creep of work time within the home also stemmed from actual increased workloads of adapting to online teaching platforms and the associated administration. The same academic explained, "I think it gave us more work in terms of having to do so much filing and so much reporting by the time it is 14:00pm, you have been doing all the admin, signing, scanning etc, that you have no time to actually get to the work of the day."

Many respondents observed the skewed distribution of this workload between male and female academics:

> In terms of the move to online teaching: all the providers of online support with whom I worked, barring one, were women. All the people working overtime, late into the night, to put together guides, were women. . . . The male colleagues I worked with . . . did not read

any of the support material, did not want to think how to adapt their assessment, made narrated powerpoints and then expected the students to manage on their own. My perception is that when something needs to be done, the men just sit on their hands while we women volunteer, without even noticing that we are the ones always doing it. (Mid-career academic, married)

One academic with young children even drew a direct parallel between the gendered nature of domestic work and the gendered nature of university work: "Women are not only carrying the burden of the nurturing of children and the housework at home, but when a crisis such as this comes along, male academics seem to get on with the important business of research, leaving women academics to nurture the students (child care) and do the admin (university housework)."

In normal times, having a workspace outside the home provided academics with a way to manage their time and energy. "Going to the office helped me put barriers between work and private life. Now these barriers are gone," said one experienced academic without children. With the spatial barriers removed, some academics found they had to institute structure to protect their time:

At home we try to follow a workday approach (9–5 type hours) at home and break to have lunch together. This also helps to separate work and personal spaces, [as] demands [for] working 'later' at home is more evident in lockdown and gets pushback from the family. (Established academic with no children)

Those who reported managing the blurring of their home spaces well, a minority, often credited their well-equipped houses with dedicated workspaces. One established academic and mother of two teenagers, who felt overworked but in control, explained her supportive environment: "I am fortunate to have a comfortable home, fibre [-optic cabling], devices and space to work." Although in this specific instance having older children with fewer dependencies was also an enabler, the challenges of blurring space and time held across academics regardless of the number and age of their children. Consequently, many yearned for a return to the physical separation of pre-pandemic times. "I can't wait to go back to work and having a clear divide between work and home," explained an established academic with no children.

Even after the state-imposed lockdown restrictions eased, institutional work-from-home mandates and fluctuant school closures made it impossible for academics to opt for the office. As work, school, and childcare activities were "squeezed under one roof", the traditional home became a physically and emotionally congested space.

**Sharing space.**   As the experiences of academics who are accustomed to working from home indicate, it was not simply performing work from one's house that presented challenges. Before the lockdown, women academics sometimes worked from home because it offered a comfortable workspace. It provided a refuge from the busy office space where colleagues or students "popped in" frequently or an opportunity to be present when children returned from school. This changed during lockdown. An established academic with a pre-schooler and a child in high school recalled the pre-lockdown situation: "I am usually very productive when working from home. It is generally quiet, and I can work in an uninterrupted space. However, during lockdown, the whole family was at home together, so there were more interruptions and more activities that had to be done."

The transformation of the home was therefore largely driven by the requirement to share space with children, partners, roommates, family members, and domestic workers in the household. Often, this required physical transformation of the house:

We have had to re-arrange our whole home environment to accommodate an actual and a virtual classroom (which means I cannot use my computer when my child has online classes); and two offices (I end up working from the spare bedroom, my partner gets the study). (Experienced academic with a child in primary school)

Some women expressed frustration with their university employer, who, as one early-career academic with no children said, "assumes that everybody has a space at home in which they could perform their normal work". For several participants, "there is no working space" that can be set aside and dedicated to work, in the words of an experienced academic with an adolescent child. Even for those without partners or children, working in the confined home space was less than ideal:

It has been very difficult for me to work from home. I live in a bachelor's flat, which is not large enough to have a dedicated workspace. During winter I normally work longer hours from the office because it is warmer and more comfortable than at home. (Early-career academic without children)

For those in smaller homes and with more dependent children, division of the space was not possible. As one early-career academic lamented, "I don't have a study and work in the same room as my baby." One experienced academic with a toddler described how "I had to work where she was playing or work in her room while she watches TV". And sometimes academics were sharing space with more than just their children, as in the experience of another mother of an infant: "I don't have a home office, so I have to work in the living room, so I share my space with my nanny and daughter during the day (a small house)".

Even if the actual size of the house was not an issue, women academics often found themselves moving throughout the house, transforming spaces intended for dining, relaxing, or sleeping into makeshift offices. For many, the workplace was not fixed within the home, but an ever-changing setup depending on the needs and demands of others in the house. Hence, the challenge of maintaining boundaries and sharing space persisted:

The biggest issue is space. Though we have a three-bedroomed house, everything–as in both my partner and my work–happened in the sitting/living room. The kids also knew they could access if burning issues/questions arose. (Established academic with one child in primary school and another in high school)

The congestion created numerous distractions, as several academics complained: "I do not have a quiet space [because] the house is fuller and noisier" (mid-career academic with a child in primary school). For those with a workspace and male partners, women sometimes felt pressure to give priority to their spouse's work by conceding the space or being the one to tend to children who breached the barriers, deprioritising their own needs for a quiet workspace. Consequently, the home became an intensive space, packed with activity and disorder. Even for those without children, this turned the home into a confrontational arena in which difficult issues had to be addressed and from which there was no escape.

**Clashes with partners.** Inevitably, after "sharing the same working space for such a long time" (experienced academic with a primary-schooler), participants found themselves clashing with the people around them, namely their partners. For some, the clash came from dealing with a partner's struggles, ailments, as in the case of one early-career academic with no children: "It is emotionally hard to be at home with a partner with depression".

In the worst-case scenario, the extended period of confinement, cohabitation, and coworking actually damaged relationships: "It brought tension in the household and in time led to a

deterioration of my relation[ship] with my husband (soon to be ex-husband)," reported an early-career academic and mother of a kindergartner.

For others, the clash came from having their workspace encroached upon. "My husband has taken over my study, so I have to work in the dining room," reported a mid-career mother of a high school student. Even when the burden of accommodating the other was mutual, the solutions for sharing space could be frustrating:

> My husband and I are both academics and while we do not have children the lockdown did impact on our academic productivity. We continue to have to work at home and space is limited. We share office space and we both need to meet with colleagues and students frequently using online platforms. We can't have meetings at the same time or if we do, one of us has to move to the kitchen. (Established academic)

Privacy, concentration, and access to resources all became more elusive under these circumstances. One established academic with a child in primary school explained, "Juggling appliances and Wi-Fi needs to meet everyone's study demands can be a challenge." As she found, the burden of sharing and meeting the needs of others was not always shared equally. "Unlike my husband, I can't just close the door and dissociate myself from the needs of the rest of the family." Many women academics with male partners and children reported similar experiences of feeling unfairly tasked with sacrificing their privacy or dedicated workspace and time to meet the needs of the family. As the early-career mother of two toddlers recalled: "I have a very involved husband [but] when we have a clash, it is the women's work that suffers."

## Impact

**Work suffers.**　When discussing their changing home environments during lockdown, most women academics reported an adverse impact on their academic output, productivity, or dedication. The issues and barriers they reported were often directly related to their environment, shared space, and pressures from family members. One early-career academic summarised how these elements could combine to interfere with work:

> I struggled without having a defined workspace of my own, as I am extremely sensitive to and affected by the emotions/moods of others, in this case, my partner. I found it difficult to create barriers between professional and personal life, without having a physical barrier (i.e. a door or wall!) between us, nor the opportunity for solitude and separation. It definitely did have an effect on my ability to focus on my own work, on my research, and creative output.

Hence, what might have been a demarcated space for work was constantly disrupted and breached by others in the home. Consequently, women expressed frustration about the expectation that "duties are supposed to be professionally performed in the domestic space, which is one of interruption and constant demand" (established academic and mother of a senior high school student).

Even if women academics had dedicated spaces, sometimes they found themselves competing with their partners for space or resources to complete their work:

> Managing space in the home, [there was] no dedicated office space for self. Some compromises were required that were understandable, but still had an effect on my academic work. For example, my husband's Teams Meetings for his work are very confidential and the 'best space' was required. (Established academic with a university-age child)

With children present, the home as workspace became seriously compromised as young children in particular saw the presence of their mother at home as signalling access and availability. One early-career academic and mother of a toddler and a kindergartner described how this dynamic "reduces concentration", leading to "insufficient time to do academic work".

Some women respond with a sense of resignation as their experience during lockdown led them to the realisation that "the workplace, and academia in this instance, is not well-geared towards working mothers." Particularly in academia, where research work is expected to be done outside of normal business hours, female academics who were confined to the home and stretched thin by competing family and work demands found "it is very difficult to find the head and physical space to focus on intellectual work" (Experienced academic with two primary school-age children).

Frustration over the loss in productivity seemed compounded by the converse experience women observed in their male colleagues, and their fears for their professional future:

> I see myself and other women falling behind our male peers I the international arena, as our productivity has been typically lower and I anticipate that we will see this impact in future invitations to participate in conferences, international research projects, and the like (Established academic with no children in the home, but caring for a parent).

For women academics trapped at home, difficult issues fill the confined space with stressful emotions. Several respondents spoke of how increased anxiety, financial difficulties, and feelings of sadness became features of their home environment, impacting their work:

> The sad loss of income and critical resources due to the numerous trade-offs one has had to make. Needless to say, this has caused more tension and conflict on the homefront, with dire consequences on academic work. (Early-career mother of a primary-schooler and a teenager)

**Mental and emotional health toll.** Ultimately, many participants discussed adverse impacts on their personal health within their newly challenging home environments. One experienced scholar described how her increased workload, coupled with the pressure to step up with household chores affected her mental health: "I did suffer from panic attacks during this time". Although a virtual health service was available through her university, she explained, "I did not feel comfortable sharing my sorrows with my workplace".

While this respondent was able to resist the pressure to do chores during their work time, thanks to the support of other family members with household duties, others did not have this option and found it impacted their morale. "I've been a bit sad, actually, to see how quickly my family casually renders me a servant, when I am constantly around and not able to avail myself of the professional workspace," explained an established academic and mother of a senior high school student.

For those at home alone, the extreme isolation led to depression and anxiety, regardless of their workspace setup. "I have settled in at working at home (office space set up really well) but I often get lonely, which makes me feel depressed," explained one academic who was single. The experience of one established academic without children was compounded by being far from her 'real' home: "One week down the line I started to feel depression setting in, and anxiety. . . . As an expat I became very lonely and homesick. This negatively impacted on my work".

Others described the negative feelings they began to develop toward the work itself. Describing her overwhelming frustration with her blurred work-home space, one established academic with no children wrote, "I simply do not experience joy in my work context at this

stage". The experience of another established academic was one of several who reported that "never leaving the house affected my motivation to work. . . . I feel battered, emotionally".

A few academics were able to work around the lockdown conditions by getting out of the home. An established academic with one child in primary school and another in high school that "I now have an away-from-home small place to stay where my new job is", which she made use of to "finalise articles". Even when academics had the luxury of this option, however, they were still burdened with feelings of anxiety and guilt about their choice to escape the home for their work, as this participant explained. "The difference is . . . invaluable. . . But I will be seen by many as the 'mother-working-away-from-home' . . . *that* mother."

For the majority confined to the home and unable to escape to the office, they were also unable to access the people and places that help them to cope with the stresses of life. For instance, as one established academic explained, "I can't sleep because my mind is thinking about so many things that I haven't done. . . . I can't go to church to re-charge emotionally. . . . Everything is just too much."

Under these conditions, home is anything but a refuge from the outside world for women academics.

**Lack of social interaction.**   It is important to address the notion that several participants lamented about missing interactions with their peers, as one academic and early-career mother of 2 children in primary school stated: "I miss my colleagues. I miss having a laugh in the passage. I miss being real with people. This remote meeting system only allows so much body language and I know that there are people who are suffering, and we are all blissfully unaware". Similarly, a mid-career academic with no children uttered a similar tone of frustration: "I am already missing my colleagues—chatting, banging hands and smelling a bright new day outside the gate. I miss my workplace. We take those informal greetings on the bus for granted. Lord hear us. We want to go back to the old normal. The new normal is not good for emotional balance". A mid-career academic without children lamented: "I am eagerly awaiting my return to my campus office. I love the academic, emotional freedom my office affords me. I miss the social interactions, the joy and comradeship at a colleague's elation, a manuscript is accepted for publication". These reflections were not random but pertinent in our dataset.

We know that travel to conferences and workshops often include sharing research and ideas, and time and again, this leads to collaboration on various research projects. It is evident that not having these interactions negatively affected female researchers' mental states and productive capabilities which one mid-career mother of 2 pre-schoolers and a child in primary school affirmed: "Also because of the lockdown, I should have travelled to 4 international conferences, which would have given me proceedings, but that was cancelled. So my research output for this year is going to be close to zero".

## Discussion

In recounting the challenges and impacts of working from home during the COVID-19 lockdown, the women academics in this study painted a picture of a home environment that changed in both dramatic and subtle ways. As one aim of this study was to establish the experience of a subset of women professionals within an African context, these findings are steeped in the unique lockdown conditions and socio-economic position of women workers and academics in South Africa and may contrast with those from other regions and professional backgrounds. Forced confinement as a precautionary measure transformed the home into a safe space for women in Turkey [56], or a place to bond with family for women in Korea [57]. But for South African women academics, increased workloads, domestic pressures, and inadequate workspaces transformed the home into a more stressful place. Their experiences, which were

shaped entirely by their forced confinement in the home and the disruption of childcare, healthcare for family members, and domestic work, provides a deeper explanation for the lost productivity and emotional tolls reported elsewhere in the literature on women academics during the COVID-19 pandemic [13, 23] as both of these negative impacts were described in great detail by the participants in this study.

## The altered home space

This study clearly highlighted the dramatic pressures on women with children, consistent with other studies which found that mothers felt extreme pressure to sacrifice their professional roles in order to uphold their parental roles while at home with their families during the lockdown [58]. However, unlike other studies which point to motherhood as being the critical factor impacting the productivity and experience of women academics [15], and that women academics with young children have been most adversely affected [59], this study shows that the negative experiences of the changes in the home were felt by most women, regardless of their parental status or the age and number of their children. The gendered nature of these negative experiences, as women felt increased pressure to assume the specific role of household leader, at the expense of being a work leader, once in the "domestic sphere", reinforces concerns raced by Mallet et al. [36] about the social meaning of spaces and the gendered meanings commonly endowed in the home. The sharing of the home space with partners–as well as extended family members–seemed to be a significant reason for this trend as most participants in the study were partnered and described the challenges of suddenly "sharing the same working space for such a long time". Hence, even without children, women found themselves feeling very physically and emotionally crowded in their home spaces.

Consistent with similar studies of women academics in which women reported that they assumed a disproportionate amount of the responsibility for household work compared to their partners [15, 23, 60–62], many participants described how they felt greater pressure to de-prioritise their work and perform household duties than their male partners. While this featured as an inconvenience for some, for others it was a wakeup call to endemic gender imbalances in both society and the academy as many wondered if their experiences were being shared by male colleagues. The clashes with partners over how to share the workspace or division of labour provided some with opportunities to problem-solve: implementing structured schedules, dividing up offices and rooms, and finding external office space. Notably, in all such situations described by women with male partners (there were no overt mentions of female partners), the change or compromise was always actively made by women academics. They were the ones to leave the room, change their schedules, or seek alternatives, while their partners "get the study".

Following from this, the biggest factor of the shared experience among women academics was the limits of the space itself: how the physical structure of their homes impeded their ability to carve out spaces that were suitable for work, family, and leisure activities. The near-ubiquitous references to "blurring of boundaries" among participants is the ultimate expression of this phenomenon and has been described elsewhere in the literature on family experiences during COVID-19 lockdown [24, 63, 64]. Without a dedicated workspace in the home, or with the invasion of the home office by other members of the household, women academics found it impossible to fulfil themselves personally and professionally. The distinction between home and office, between personal time and work time, becoming all but invisible.

## An altered sense of place

As physical boundaries in the home blurred, so too did the emotional boundaries for women academics. Not only did they begin to feel anxiety from increased challenge of juggling

numerous responsibilities at once, but they also felt their professional motivation and satisfaction depleting under the confined conditions. Similar feelings of inadequacy and loss of motivation amidst the lockdown challenges were found in similar studies in Italy [24] and Turkey [65]. Comments from participants who had previously worked from home as part of their normal routine, and who had done so gladly and with great success, suggested that these emotional setbacks were not simply linked to working within the home place. Rather, it was the forced confinement, the removal of options to seek external help, the social isolation, and increase in workload that transformed the home into a place that was no longer comforting as a workspace or a living space. Under these conditions, many found that the reality of gender became almost unavoidable, rendering them responsible for all household needs first (homeschooling, chores, emotional support, caretaking of children and family members), and professional needs last.

In a situation where the lockdown had "squeezed everything under one roof", many described how they simply postponed any professional goals that were not absolutely 'essential' (i.e., teaching duties), suspending their studies, research, papers, and proposals in order to take charge of household duties. As women deprioritised their academic work in this way, or alternatively attempted to meet every personal and professional goal, many found themselves either burnt out or stinging from a sense of failure to 'do it all'. In either case, in the home women were subject to the "double burden" of being a worker and a household manager [35], and reinforcing gender stereotypes about 'women's work' and the value of women within their organisations. Importantly, this sense of women's contributions in both the house/family and the professional field being undervalued lived within women as their sense of place blurred. While many in the study detailed the extreme lengths they went to in order to meet the deadline, please their families, maintain their homes, and objectively perform, they often internalised the chaos created within their home spaces, citing it as evidence that they had not, in fact, succeeded. In the absence support mechanisms outside of the home–through extended family, schools, employers, or community groups–women labour under the double burden, even when they appear to be strong and coping. As Mallett et al. [36] suggest in their own critique of homebased work, if universities, and employers more generally, endeavour to nurture and retain talented women, they cannot afford to overlook the gendered nature of the home space for their employees, and its impacts for support systems.

Examining the implications for altered spaces are crucial for ensuring women's academic output is not disproportionately affected by the enforced pandemic lockdown; and in that way safeguarding the career trajectories of female academics. The gendered nature of childcare and domestic tasks as well as single-parent households, the majority of which are female-headed, have drastically affected female academics and their futures in the academy.

## Conclusion

This study has shown that for women academics in South Africa, the meaning of home was completely unsettled by the pandemic. Under lockdown conditions, home is no longer that place of refuge from the demands and disorder of the outside world. It is certainly not a neat-and-tidy space in which the only dualism to resolve is that of the traditional office space versus home space. The home became a crowded and congested place in which many other functions–from home schooling to online teaching to new administrative tasks–needed to be executed, sometimes without the help provided by otherwise reliable partners, domestic workers, or extended family members.

It is too early to tell how this transformation of the home will persist for women academics going forward as lockdown conditions are in flux and the world continues to on its journey

toward the 'new normal' in extremely uncertain conditions. Nor is it clear how future strains of the coronavirus, unforeseen epidemics, or reimagined education systems might impact on 'work from home' arrangements into the future, such as is anticipated in Maloney and Kim's [66] recent work on "the low density university". In light of this uncertainty, we argue that future theorising on the meaning of home for women academics, and its implications for work from home models, must take account of the several elements.

First, in the future, home is likely to be a congested space in which there is competition for limited resources (space, broadband, time). The outcomes of working in that space are likely to continue impacting negatively on women's academic work, and their future career prospects. The first step for institutions toward establishing the right structures to manage the gendered nature of home workspaces is to recognise this as a talent management issue [67], one which they must prioritise. Accordingly, institutions must be explicit in setting a goal of maintaining the career progression of women academics into senior roles and creating supportive policies and an enabling culture.

Second, the precise meaning of home and what it means to work from home for women academics will be different for single mothers with children; older mothers with grownup children; women without children; mothers with partners and children in the home; single women; and women with toddlers and small children, etc. These differences among academic women working from home must be teased out in much finer detail in empirical and theoretical work.

Third, that the meaning of home has to take account not only of academic productivity but also its relation to the social, emotional, and overall health constraints and challenges of working under confinement. The work of women academics is inescapably influenced by the extent to which they experience a sense of well-being in the domestic workspace.

Fourth, that the meaning of home cannot be explained outside of the gender inequalities that divide the allocation and understanding of 'work' between women and their partners [18, 68]. Once in the home, most women academics will have to contend with these inequalities in expectations and obligations in ways that they would not otherwise in a professional setting. This should not stop women from demanding practices and processes that acknowledge the gendered nature of home spaces while also pursuing challenging projects that enhance their careers [69]. This means also avoiding decisions or projects that keep them in support or follower roles [70].

Future theoretical work will have to account for the micropolitics of the changing meanings of home and how exactly space is claimed, conceded, and constantly negotiated between academic women and their partners, as well as the children and other occupants under the same roof. In this way, under work from home policies, academic institutions are no longer interfacing with their employees alone, but all members of the employees' household. Models for how these relationships are enacted and managed must be conceptualised. Thus, clear policy directions for higher education institutions to remedy the negative consequences of the pandemic lockdown for women's academic work at home and their professional futures need to seriously be taken into account. There is no doubt that science and innovation benefit from diversity. We need inclusive universities. To this end, the implications that the loss of women's scientific expertise to the academy needs to be seriously considered.

## Author Contributions

**Conceptualization:** Cyrill Walters, Linda Ronnie, Jonathan Jansen, Samantha Kriger.

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
