## [Decision Letter · Decision Letter 0]

5 Sep 2022

PONE-D-22-15619The changing meaning of home in the work of South African women academics during the pandemic-enforced lockdownPLOS ONE

Dear Dr. Walters,

Thank you for submitting your manuscript to PLOS ONE. After careful consideration, we feel that it has merit but does not fully meet PLOS ONE’s publication criteria as it currently stands. Therefore, we invite you to submit a revised version of the manuscript that addresses the points raised during the review process.Specifically, please provide more details on the data collection strategies and specify if the data is available. In addition, please discuss other confounding factors that may affect female researchers' work performance.

We look forward to receiving your revised manuscript.

Kind regards,

Jianhong Zhou

Staff Editor

PLOS ONE

Journal Requirements:

2. Please provide additional details regarding participant consent. In the Methods section, please ensure that you have specified (1) whether consent was informed and (2) what type you obtained (for instance, written or verbal). If your study included minors, state whether you obtained consent from parents or guardians. If the need for consent was waived by the ethics committee, please include this information

3. We note that you have stated that you will provide repository information for your data at acceptance. Should your manuscript be accepted for publication, we will hold it until you provide the relevant accession numbers or DOIs necessary to access your data. If you wish to make changes to your Data Availability statement, please describe these changes in your cover letter and we will update your Data Availability statement to reflect the information you provide

4. Please ensure that you include a title page within your main document. You should list all authors and all affiliations as per our author instructions and clearly indicate the corresponding author

Reviewers' comments:

Reviewer's Responses to Questions

**Comments to the Author**

1. Is the manuscript technically sound, and do the data support the conclusions?

Reviewer #1: Yes

2. Has the statistical analysis been performed appropriately and rigorously? 

Reviewer #1: N/A

3. Have the authors made all data underlying the findings in their manuscript fully available?

Reviewer #1: No

4. Is the manuscript presented in an intelligible fashion and written in standard English?

Reviewer #1: Yes

5. Review Comments to the Author

Reviewer #1: Thank you for the opportunity to read and comment on your research.

The research article investigates how the meaning of home and working from home changed during COVID-times for women academics. The dataset includes close to 2000 observations derived from online surveys. The authors find that home remains an unequal place for women's work, and working from home during the pandemic had dire consequences for academic careers.

Comments:

I think the authors should emphasise the problem that the concept of "home" has changed for women academics. They mention that the meaning of home has changed from being a place of refuge to a place congested, competitive and constrained. However, they do not sufficiently highlight the implications of women not having a refuge or the complications and threats women academics face for not matching their male counterparts' performance.

I am also wondering about the relevance of the research for the future. What do these results imply for women academics in the future, and how can policies be implemented to support women academics who have to perform and prove themselves?

Specific comments:

Data: the authors should add descriptive statistics related to the sample demographics. They mention that the sample includes different levels of academia, but they do not provide frequencies (thus, how many from the sample were from which level). They also do not indicate the age, race, household size or institution distributions. Therefore they keep the reader wondering who the respondents were and if the sample is representative of female South African academics. There is most likely a selection bias in that female academics that were active and managed to adapt to the "new normal" did not take part in the survey as they did not have the time. The authors should acknowledge data limitations.

Furthermore, it seems that the data is not available to the readers. Thus, the reader cannot answer questions by exploring the data. I believe adding demographic variables can result in a more in-depth analysis. From an "intervention" point of view, this is especially important to determine which female academics suffered the most.

Interestingly, in the narrative, I see no mention of the academia missing the interaction and conversation with their peers. Academics travel to conferences and workshops, share their research and ideas, and collaborate on various research projects. I believe that not having these interactions also negatively affected female researchers' mental states and productive capabilities.

6. PLOS authors have the option to publish the peer review history of their article (what does this mean?). If published, this will include your full peer review and any attached files.

Reviewer #1: No

---

## [Author Response · Author response to Decision Letter 0]

17 Nov 2022

A letter of response has been attached.

---

## [Editor Report · Decision Letter 1]

22 Dec 2022

The changing meaning of home in the work of South African women academics during the pandemic-enforced lockdown

PONE-D-22-15619R1

Dear Dr. Cyrill Walters,

We’re pleased to inform you that your manuscript has been judged scientifically suitable for publication and will be formally accepted for publication once it meets all outstanding technical requirements.

Kind regards,

Miwako Hosoda

Academic Editor

PLOS ONE

Additional Editor Comments :

This paper reveals how the meaning of home has been transformed in the face of forced " working from home" in a corona pandemic situation, with a particular focus on female researchers. An online survey of more than 2,000 female researchers is conducted. An online survey of more than 2,000 women researchers was conducted. And it was empirically shown that the meaning of home has changed from being a refuge from the outside world to a gendered space that is crowded, competitive, and constrained. The home is a terribly unequal place for working women, with disastrous consequences for their academic careers. To change this situation, the paper suggests that a deeper understanding of the social meaning of home and the implications of the "new normal" of telecommuting is needed. This is an issue that is shared not only by South African women researchers but also by women researchers around the world, and I think it is very significant to continue researching the theme of this paper.

This paper seems to have improved on the problems with the reviewers' input to date. Therefore, since it is also important to publish at the appropriate time, I would like to approve it as Accepted.

---

## [Editor Report · Acceptance letter]

2 Jan 2023

PONE-D-22-15619R1 

The changing meaning of ‘home’ in the work of South African women academics
during the pandemic-enforced lockdown 

Dear Dr. Walters:

I'm pleased to inform you that your manuscript has been deemed suitable for publication in PLOS ONE. Congratulations! Your manuscript is now with our production department. 

Kind regards, 

on behalf of

Dr. Miwako Hosoda 

Academic Editor

PLOS ONE